# Cognitive-Motor Training Improves Reading-Related Executive Functions: A Randomized Clinical Trial Study in Dyslexia

**DOI:** 10.3390/brainsci14020127

**Published:** 2024-01-25

**Authors:** Mehdi Ramezani, Angela J. Fawcett

**Affiliations:** 1Nursing and Midwifery Care Research Center, Health Management Research Institute, Iran University of Medical Sciences, Tehran 14496-14535, Iran; mramezaniiiiii@gmail.com; 2Department of Psychology, Swansea University, Swansea SA1 8EN, UK

**Keywords:** dual task, single task, cognitive-motor training, executive function, reading, dyslexia, cerebellum, clinical trial study

## Abstract

Children with developmental dyslexia (DD) often struggle with executive function difficulties which can continue into adulthood if not addressed. This double-blinded randomized clinical trial study evaluated the short-term effects of the Verbal Working Memory-Balance (VWM-B) program on reading-related executive functions, reading skills, and reading comprehension in Persian children with DD. The active control group [12 children with DD with a mean age of 9 years (SD = 0.90)] received training using the single-task VWM program, while the experiment group [15 children with DD with a mean age of 8 years (SD = 0.74)] received training with the dual-task VWM-B program. Both groups received fifteen training sessions, and assessments were conducted before and after the intervention. The groups were homogenized for possible confounders of age, gender, IQ level, and attention level. The study employed separate mixed ANOVA analyses to estimate the impact of training programs on various measured functions. Significant improvements were observed in the outcome measures of backward digit span, text comprehension, verbal fluency, Stroop color–word test and interference, and the reading subtests. Additionally, significant correlations were found between reading skills and backward digit span, text comprehension, verbal fluency, and Stroop variables. In conclusion, the dual-task VWM-B program was found to be more effective than the single-task VWM program in improving selective attention, cognitive inhibition, verbal working memory capacity, information processing speed, naming ability, and lexical access speed. These enhanced executive functions were associated with improved reading skills in children with DD.

## 1. Introduction

In a comprehensive definition, developmental dyslexia (DD) refers to slow and inaccurate word recognition, which causes impairments in learning to read fluently and accurately [1]. However, children with DD have sufficient intelligence, experience conventional classroom conditions, and access suitable socio-economic opportunities [2]. The prevalence of DD in elementary school students is considerable (5–17.5%) [3]. Therefore, providing therapeutic interventions to improve reading ability would be valuable for this population.

In recent decades, numerous studies have been conducted on dyslexia to explain the symptoms of this condition and find ways to improve reading skills. Individuals with DD exhibit slower and less accurate motor programming and sensory information processing due to inefficiencies in error correction (defective motor-perceptual function) [4,5]. They often struggle with limited motor and balance skills automation, requiring conscious compensation, which can lead to deficits in reading and other cognitive functions [6,7]. DD is not just a result of deficient automatization, but rather the automatization of abnormally developed functional coordination, known as functional coordination deficit [8]. This deficit in functional coordination between grapheme and phonological letter representations is observed in children with DD [9]. The literature highlights the impaired visual and auditory mechanisms that cause phonological problems in DD [10,11]. Children with DD have difficulties reading words and nonwords due to impaired sound reception and discrimination [12] and struggle to distinguish between the visual processing of linguistic and non-linguistic materials [13]. All these factors disrupt cognitive functions in individuals with DD, and several studies have acknowledged the existence of issues with cognitive and high-level cognitive processes [7,14,15,16]. It is important to note the presence of problems with executive function(s) (EF) [17,18,19,20,21,22,23].

There are many different definitions for EF, but in the most accepted definition, EF is an umbrella term for a set of complex and high-level cognitive processes, such as attention, inhibition, flexibility, and working memory (WM) that conduct flexible and goal-directed behaviors [20]. Over the past decades, the importance of EF for learning to read has become more obvious. Children with DD exhibit deficits in EF [24] which may persist into adulthood if left unremedied [23]. One of the basic features of EF is attention [19], and one type of attention is selective attention which is defined as the cognitive process of attending to important information (external or internal sensory stimuli) while suppressing or ignoring other unimportant and distracting stimuli [25]. Although attention disorders in DD are not as severe as in attention deficit hyperactivity disorder (ADHD), some reports show deficits of selective attention in DD [15,26]. Inhibition of irrelevant and distracting information processing seems to be a crucial function of selective attention [27]. Cognitive inhibition is the basic EF and plays a critical role in preceding the development of other EFs [28]. Inhibitory control is usually described as the ability to suppress cognitive processes that can cause interference [28]. Some populations with cognitive deficits are usually unable to inhibit certain kinds of irrelevant and distracting information [27]. Research has confirmed the existence of deficits in the inhibitory control of cognition in DD [28,29,30]. In children with DD, deficits in cognitive inhibition cause difficulties in word recognition [29]. If a deficient cognitive inhibition ability appears concurrently with WM dysfunction, these children have more difficulties in word recognition [29].

The limited-capacity system of WM is an EF defined as the ability to temporarily store, process, maintain, integrate, and manipulate information from various sources [31]. WM is divided into three components: a central executive component, which is a supervisory control system that has limited attentional capacity, and two slave components that are responsible for phonologically-based information (the phonological loop) and visual and spatial information (the visuospatial sketchpad) [31]. The phonological loop is also called verbal WM [31]. A growing body of literature confirms verbal WM dysfunction in DD [32,33,34]. The possibility of a deficit in the central executive component of WM in dyslexia has also been raised [32,33,35]. The connection between the DD and the deficient visuospatial component of WM is controversial, with evidence either to confirm or ignore it [33,36]. It seems that if WM capacity does not increase in DD, it could extend into adulthood and affect performance in all components of the WM [32].

After releasing the double-deficit theory [37], researchers paid more attention to processing speed, and many studies have shown deficits in information processing speed in DD [38,39,40,41]. Processing speed refers to the number of accurate responses a person can generate in a task within a given timeframe, utilizing various cognitive functions such as motor and visual scanning speed [42]. A decrease in information processing speed could negatively affect other cognitive functions, as processing speed mediates other cognitive domains such as selective attention [43], WM [44,45], naming speed [41,46,47], and verbal fluency [43,48]. As mentioned, selective attention and WM are necessary for learning to read. Moreover, difficulties in naming speed and verbal fluency have been associated with DD [49,50,51]. Deficits in precise timing mechanisms could inhibit the ability of some individuals with DD to conduct rapid naming processing [52]. Since naming speed is correlated with a range of reading skills, and this deficit might persist into middle childhood, early intervention could be helpful [49,50,52]. Moreover, improving information processing speed in DD is necessary because it may persist into adulthood and negatively affect other EFs [39].

As DD is a complex and multifactorial condition [53], it has been recommended that cognitive and motor aspects should be integrated in training individuals with DD. Developing and employing appropriate interventions to improve reading-related EFs (selective attention, inhibition, WM, processing speed, naming speed, and verbal fluency) in children with DD appears necessary [54]. Moreover, balance and postural control difficulties in dyslexia cannot be overlooked. Sensorimotor and postural training may enhance attention, coordination, and postural stability by boosting cerebellum activation [55,56]. Among computer-based training programs evaluated in DD, some positive effects have been shown on EF [22,57,58,59]. These training programs were usually single-task; however, it has been suggested that training by two or more modalities in combination would be more effective [2,22,60]. Dual-task training protocols can effectively modulate attention, EFs, and standing postural control in different populations [61,62,63]. Based on our best knowledge, the only computer-based dual-task (a mix of cognitive and balance-related performance) training program, called Verbal Working Memory-BalanceVWM-B, has recently been evaluated in DD [2,60]. The positive effects of the VWM-B have been shown on the cognitive (verbal WM capacity and reading skills) and motor functions of children with DD [2,60]. It is widely admitted that WM plays a crucial role in DD [2,32,34,36]. Furthermore, the VWM-B program has been shown to improve verbal WM capacity [2,60]. Therefore, it is reasonable to assume that VWM-B may also improve other EFs related to reading.

The study aimed to evaluate the short-term effects of the dual-task VWM-B program training on reading-related EFs (selective attention, inhibition, WM, processing speed, naming ability, and verbal fluency), reading skills, and reading comprehension in Persian children with DD aged 8 to 10. In essence, the study compared the progress of children with DD in two groups. One group received a WM battery using the single-task Verbal Working Memory_VWM program (active control group). In contrast, the other group received the same WM battery under dual-task balance conditions (experimental group). The study discussed the potential effects of the VWM-B program on reading-related EFs, reading skills, and reading comprehension using various behavioral outcome measures.

## 2. Materials and Methods

### 2.1. Subjects and Design

This study is a quasi-double-blind randomized clinical trial that includes both between-subjects and within-subjects factors. The between-subjects factor consisted of two groups, an active control group and an experimental group, while the within-subjects factor included two rounds of measurements conducted before and after the intervention. The data for the study were collected from children with previously diagnosed DD who were between 8 to 10 years old and attending public elementary school in the second to fourth grades, and this was completed at the Dyslexia Rehabilitation Center in District 20 Education Office, Tehran, Iran. As shown in Figure 1, a total of 30 children with DD who met the inclusion criteria participated in before-intervention assessments. After the intervention, however, data from 27 children with DD, with an approximate dropout rate of 10%, were analyzed. All 27 children were assigned to either the active control group or the experimental group, with the active control group consisting of 12 children with DD and the experimental group containing 15 children with DD. The study sample size was consistent with past comparable studies [2,60]. A post-hoc power analysis was also conducted to show the power of the current study sample size. The analysis indicated that, with a significance level of 0.05, the total sample size of the current study (N = 27) had 80% power, which is sufficient for the study’s purposes. Figure 1 illustrates the recruitment process for the study samples.

As shown in Figure 1, all participants (N = 30) were allocated into two equal groups by a block randomization method using Excel (2013) software [64]. First, we wrote participants’ names on separate pieces of paper and placed them in a bag. The papers were drawn randomly to assign each person a number from 1 to 30, which was then encoded in a column in Excel. Next, we divided participants into five blocks of six people. The first six people were numbered by block 1, the second six by block 2, and so on, with the last six numbered by block 5. Then Excel’s Randbetween function was used to randomly divide each block into three participants for the control group and three for the experimental group. We repeated this process for all five blocks, resulting in 15 participants in each group.

The study groups were homogenized for possible confounders of age, gender, intelligence quotient (IQ), and attention level. To estimate IQ levels, the Wechsler Intelligence Scale for Children Fourth Edition (WISC-IV) was used in the preliminary screening step [65]. The attention level was estimated using the Persian version of the Child Symptoms Inventory Parent Checklist (CSI-4) items 1 to 18 out of 97 [66]. All participants were children previously diagnosed with DD, but additional assessments were conducted to confirm the diagnosis. For this, the word reading efficiency subtest (WRT) and the non-word reading efficiency subtest (NWRT) of the Persian battery of reading tests—NEMA—were used [67]. Children were diagnosed with DD if their WRT and NWRT scores were 25% or less of the total scores [2,67]. The baseline scores of the clinical measures were collected in the before-intervention step. The WRT and NWRT scores obtained in the screening step were used as the before-intervention scores for children recruited for the study [2]. Both groups underwent the intervention step, which consisted of fifteen sessions in five weeks, three days per week, one session per day, and 45 to 60 min per session. The active control group received training through the single-task VWM program, while the experimental group received training through the dual-task VWM-B program. Finally, after-intervention assessments were conducted with a mean (SD) of 43 (6.94) days between the before-intervention and after-intervention assessments.

Participants were included based on the following criteria: normal IQ level (WISC-IV total score ≥ 85), normal attention level (CSI-4 1–18 items total score ≤ 6), normal or corrected vision and hearing conditions, native-Persian language, right-handedness (tested by Edinburgh handedness inventory), and average socio-economic status (reported by the families) [2,60,68]. Individuals with a history of neurological or psychiatric disorders and those using drugs that affect the central nervous system were excluded. Data from participants who did not attend at least 75% of the entire intervention sessions (i.e., 12 out of 15) as well as those who did not participate in the after-intervention assessments were excluded from the analysis [2,60].

In the current study, the participants and an evaluator of the before and after-intervention assessments were blinded to the group allocation. The evaluator was an independent individual not part of the research team. Despite blinding the participants, they may have noticed differences between the training programs, which could have led them to identify which type of training they received. Therefore, the study design could be considered quasi-double-blind [2,60].

### 2.2. Study Measures

The study utilized a variety of questionnaires to measure various variables, including reading skills, comprehension, and reading-related EFs, such as selective attention, WM capacity, inhibition, information processing speed, naming ability, and verbal fluency (lexical access). The questionnaires used in the study included the WISC-IV, CSI-4, NEMA reading subtests, the Edinburgh handedness inventory, the backward digit span (BDS), the trail-making test part A (TMT-A), phonemic and semantic verbal fluency tests (PVFT and SVFT), a text comprehension test (TCT), and Stroop subtests.

To ensure that the participants met the inclusion criteria, some of these questionnaires were used in the preliminary screening step of the study. The WISC-IV, CSI-4, two subtests of the NEMA (WRT and NWRT), and the Edinburgh handedness inventory were all validated scales used in the screening process. The total score of the WISC-IV was calculated to estimate the participants’ IQ level [65], and children with a score of <85 were excluded from the study since mental disabilities can lead to learning disabilities [2,69]. The parent checklist of the CSI-4 was used to estimate the participants’ attention level [66], and children with total scores of ≥7 were excluded from the study since DD has comorbidity with ADHD [60,70]. The WRT and NWRT subtests of the NEMA were used to verify the existence of DD in the recruited participants since the ability to read words and nonwords is crucial in diagnosing DD for those following the phonological deficit theory [2,71]. The data obtained for the WRT and NWRT subtests of the NEMA in the screening measurements were used as the before-intervention score for recruited participants. The Edinburgh handedness inventory was also utilized to verify right-handedness in the recruited participants [68].

In addition to the WRT and NWRT, the chain word (CWT) and phoneme deletion (PDT) subtests of the NEMA were used to assess the changes in participants’ reading skills in both groups before and after the intervention. The BDS, which involves the phonological loop and the central executive system in Baddeley’s WM model, was used to assess the changes in the verbal WM capacity and central executive before and after the intervention [31,72,73]. TMT-A was utilized to show the changes in cognitive and visuomotor processing speed before and after the intervention [74,75]. Additionally, PVFT and SVFT were employed to assess the change in different aspects of the verbal fluency function, including phonemic and semantic fluency, before and after the intervention. Verbal fluency is known as an EF that usually requires cognitive functions such as inhibition, vocabulary size, lexical access speed, and WM [76,77,78]. However, in the present study, any alteration to the lexical access speed was estimated and interpreted by PVFT [79]. TCT was used to measure the changes in reading comprehension before and after the intervention. Previous studies have indicated the association of reading comprehension with WM, cognitive inhibition, and processing speed [17,21,80,81,82]. The Stroop test was used to measure processing speed, selective attention, inhibition, and naming ability [2,60,83,84]. It includes three color naming, word naming, and color–word components [2]. Participants were required to name the color of all 176 bars in the Stroop color naming subtest (SCT). All the bars were colored red, blue, green, or yellow. In the Stroop word naming subtest (SWT), participants read 176 terms printed in different colors and skipped the color of the words. In the Stroop color–word subtest (SCWT), they named the color of 176 words presented in the SWT by ignoring their printed form. The Stroop color–word interference (SCWI) was calculated for each participant as the time of the SCWT minus the time of the SWT [2].

### 2.3. Training Programs

As stated, the study groups received five weeks of training for 15 sessions. Participants in the active control group received single-task training through the VWM program. Participants in the experimental group received cognitive–motor dual-task training via the VWM-B program. The core structure of the WM task in both the VWM and VWM-B programs was designed with inspiration from Baddeley’s WM model [2,31]. The WM task of these programs included the encoding, maintenance, and retrieval sub-processes of verbal WM [31]. The VWM program, as a single-task program, is structurally designed to improve verbal WM capacity; however, the VWM-B program, as a dual-task program, is structurally designed to train verbal WM and balance movements concurrently [2]. In other words, the main structural difference between these programs is the balance task that the VWM-B has mixed into the maintenance and retrieval subprocesses of the VWM program [2]. The VWM training was performed by software that runs via a computerized 19-inch touch-screen monitor and a speaker to recite the words [60]. For VWM program training, the subject sat on a chair in a comfortable state, arms on the table [60]. The VWM-B program software runs using a portable robotic device [2]. As with the VWM program, the software of the VWM-B runs using a computerized 19-inch touch-screen monitor and a speaker to recite the words. In addition, the robotic device consists of a programmed platform [2]. The platform was programmed to carry out any desired tilting motion. Tilting motions made in a range of 0–20° in both mediolateral and anteroposterior or in a combination of both [2]. A force plate (sampling frequency of 100 Hz and accuracy of ±0.4 mm) on top of the platform shows the participant’s center of pressure (CoP) status on the monitor [2]. When training with the VWM-B, the participant stands on the platform, feet on the force plate with an approximate 10 cm distance between the feet, and watches the monitor with an approximate 50 cm distance at eye level [2]. It is important to note that calibration of the CoP amplitude (for the limit of stability of each participant) was conducted for safety [85]. The training phase began after the participants received clear instructions. For more details about the VWM and VWM-B programs, see Ramezani et al., 2021 [2].

In both the VWM and VWM-B programs, a training trial launches after touching the start button on the monitor. Following a three-second delay, the monitor displays the encoding sub-process of WM for ten seconds in the form of a target box. The target box contains a statement, a series of words, or a single word. Simultaneously, the pre-recorded voice recites the content of the target box. In the maintenance sub-process of WM in the VWM program, the content of the target box appears on the monitor for ten seconds through the separated component boxes.

The VWM and VWM-B programs differ in their maintenance and retrieval sub-processes of WM. Unlike the VWM program, the VWM-B includes an additional balance task that makes it dual-tasking. In the maintenance step of the VWM-B program, the monitor displays a red circle (CoP marker) and a box as a start position box. This step of the VWM-B consists of two balance forms: active and passive states. In the passive state of balance, the CoP marker and the platform underneath the participant’s feet concurrently move from the start box to each component box at the same speed and direction. After a ten-second pause, the CoP marker and platform return to the start position and the procedure repeats for all components in the correct order. In the active state, the participant actively moves the CoP marker from the start box to each component box (and vice versa) in the correct order, and the platform underneath the participant’s feet has no tilting motion (fixed). In the passive state, the participant has ten seconds to read the content of each component box when the CoP marker hits a component box. In the active state, the participant has free time to read the content of each component box when the CoP marker hits a component box.

Ultimately, in the retrieving step of both programs, twice as many boxes, including the new and practiced content, are shown on the monitor for ten seconds. In the VWM program, the user touches the box on the monitor to accept or reject the target. However, in the VWM-B program, the user has to move the CoP marker to select the correct target.

### 2.4. Analysis

The study employed a range of statistical analyses to investigate the group differences at baseline scores. The normal distribution of the variables was calculated using the Shapiro–Wilk test, and based on the distribution of the numerical variables, normal or non-normal, the *t*-test, Mann–Whitney U-test, and Wilcoxon test were used. Additionally, the group differences in categorical data were estimated using the Chi-square test with α = 0.05. Qualitative variables were reported in terms of absolute frequency (%), while quantitative variables were reported as the mean (SD).

When study groups receive an intervention, the mixed between–within ANOVA is used to verify the intervention effects over time [86], as was the case in this study. The mixed ANOVA time×group interactions (*p* < 0.05) and the effect sizes (partial eta squared ηp^2^) were reported. The effects of training programs on scores in the BDS, TMT-A, TCT, PVFT, SVFT, Stroop subtests, and the NEMA reading subtests were estimated using separate ANOVA analyses. An ηp^2^ of <0.2 shows a small effect size, 0.2 to 0.49 a medium effect size, and ≥0.5 a large effect size [87]. Post hoc tests were used to uncover specific differences between three or more group means when an ANOVA F test is significant [88]. Since the present study included two groups, the mean (SD) scores of all measures at measurements before and after the intervention were reported to show the direction of outcomes. The gain scores of the entire sample, the difference between scores in the before- and after-intervention assessments, were calculated for all the measured functions. Pearson’s correlations were then reported to explore relationships between the measured reading skills and EFs, between the measured EFs, and between the measured reading skills. Coefficient values of 0.00 to 0.34, 0.35 to 0.50, and 0.50 to 1.0 were interpreted as weak, moderate, and strong correlations, respectively [85,89].

Data analysis was conducted using SPSS 22, while Gpower 3.1 software was utilized to analyze the sample size power of the current study [90,91].

## 3. Results

The active control group consisted of 12 children with DD with a mean age of 9 years (SD = 0.90), while the experimental group included 15 children with DD with a mean age of 8 years (SD = 0.74). The age difference between the groups was not significant (*p* > 0.05). Moreover, both groups were homogenized for other possible confounding variables such as gender, IQ level, and attention level (*p* > 0.05). More detailed demographic information on both groups is shown in Table 1.

The baseline scores of the clinical measures and the changes in the scores after the intervention in the control and experimental groups are presented in Figure 2, Figure 3, Figure 4 and Figure 5. The groups had no differences in the baseline scores of the measured functions (*p* > 0.05). Figure 2a–c shows the changes in BDS, TMT-A, and TCT scores in both groups after the intervention. In Figure 3a,b, see the before- and after-intervention scores of the PVFT and SVFT in the control and experimental groups. Additionally, Figure 4a–d displays the baseline and after-intervention scores of the Stroop subtests (SCT, SWT, SCWT, and SCWI) for both groups, while Figure 5a–d shows the before- and after-intervention scores of the NEMA reading subtests (WRT, NWRT, CWT, and PDT) for both groups.

The study utilized separate mixed ANOVA analyses to estimate the impact of two training programs on various functions, including the BDS, TMT-A, TCT, PVFT, SVFT, Stroop subtests, and NEMA subtests. The analysis revealed that the ‘time’ main effect was significant for all outcome measures except the SCT. This suggests that the intervention had an impact on the scores regardless of the group allocation. However, the ‘group’ main effect was not significant for all outcome measures except the WRT, indicating that the scores of these measures did not change significantly across groups, regardless of the time effect.

Furthermore, the time×group interaction was significant for the outcome measures of BDS, TCT, PVFT, SCWT, SCWI, and all the NEMA subtests (WRT, NWRT, CWT, and PDT). These results indicate that there were significant differences over time between the two groups’ scores for the mentioned measures. In other words, both the VWM and VWM-B programs were effective in improving the functions mentioned earlier after the intervention. However, the VWM-B program was significantly more effective than the VWM program based on the mean (SD) results in Figure 2, Figure 3, Figure 4 and Figure 5. For details, see Table 2, which presents the mixed ANOVA outcomes.

Pearson’s correlation coefficients of the entire sample are reported in Table 3, Table 4 and Table 5. Table 3 shows Pearson’s correlation results between the NEMA reading subtests and the measured EFs. Positive correlations were observed between reading skills and EFs as measured by PVFT, TCT, BDS, and SVFT (*p* < 0.05). However, significant negative correlations were found between the reading skills and Stroop subtest tests (SCWI and SCWT) (*p* < 0.05). Notably, there was no significant correlation between reading skills and TMT-A (*p* > 0.05).

Furthermore, additional Pearson’s correlation analyses were conducted to investigate the associations between measured EFs, including BDS, TMT-A, TCT, PVFT, SVFT, SCT, SWT, SCWT, and SCWI. As shown in Table 4, Stroop tests had significant correlations with SVFT and TCT (*p* < 0.05). Also, TCT was correlated with BDS. BDS was correlated with PVFT (*p* < 0.05). Table 4 displays significant correlations among Stroop tests (SCWT with SCWI and SCT; SCT with SWT) (*p* < 0.05).

**Table 4 brainsci-14-00127-t004:** Pearson’s correlation between the outcomes of executive functions r (*p*-value).

Outcomes	TMT-A	TCT	PVFT	SVFT	SCT	SWT	SCWT	SCWI
BDS	0.23 (0.241)	**0.42 * (0.030)**	**0.39 * (0.046)**	0.30 (0.124)	0.30 (0.882)	0.02 (0.941)	−0.24 (0.235)	−0.25 (0.200)
TMT-A	-	0.33 (0.241)	0.18 (0.367)	0.23 (0.243)	0.02 (0.894)	0.20 (0.307)	0.16 (0.423)	0.07 (0.745)
TCT	-	-	0.36 (0.062)	0.26 (0.183)	−0.08 (0.697)	−0.02 (0.916)	**−0.49 * (0.010)**	**0.50 * (0.008)**
PVFT	-	-	-	0.28 (0.157)	0.00 (0.985)	0.29 (0.149)	−0.02 (0.907)	−0.17 (0.403)
SVFT	-	-	-	-	−0.10 (0.604)	0.21 (0.292)	**−0.39 * (0.049)**	**0.51 ** (0.007)**
SCT	-	-	-	-	-	**0.70 ** (<0.001)**	**0.54 ** (0.004)**	0.21 (0.289)
SWT	-	-	-	-	-	-	0.33 (0.095)	−0.16 (0.430)
SCWT	-	-	-	-	-	-	-	**0.88 ** (<0.001)**

Note: Bolded values indicate statistically significant *p*-values (*p* < 0.05). * Correlation is significant at the 0.05 level (2-tailed). ** Correlation is significant at the 0.01 level (2-tailed). Abbreviations: TMT-A, trail-making test part A; TCT, text comprehension test; PVFT, phonemic verbal fluency test; SVFT, semantic verbal fluency test; SCT, Stroop color test; SWT, Stroop word test; SCWT, Stroop color–word test; SCWI, Stroop color–word interference; BDS, backward digit span.

Finally, Pearson’s correlation analyses were conducted to show the associations between different NEMA reading subtests. Table 5 presents the results and highlights the statistically significant correlations among the subtests. Specifically, WRT showed significant correlations with NWRT, CWT, and PDT. NWRT showed a significant correlation with CWT. Similarly, PDT showed a significant correlation with CWT (*p* < 0.05).

**Table 5 brainsci-14-00127-t005:** Pearson’s correlation between the measured reading skills r (*p*-value).

Outcomes	NWRT	CWT	PDT
WRT	**0.39 * (0.048)**	**0.72 ** (** **<** **0.001)**	**0.46 * (0.017)**
NWRT	-	**0.45 * (0.018)**	0.34 (0.086)
CWT	-	-	**0.44 * (0.021)**

Note: Bolded values indicate statistically significant *p*-values (*p* < 0.05). * Correlation is significant at the 0.05 level (2-tailed). ** Correlation is significant at the 0.01 level (2-tailed). Abbreviations: NWRT, nonword reading test; CWT, chains word test; PDT, phoneme deletion test; WRT, word reading test.

## 4. Discussion

The study aimed to evaluate the effectiveness of the dual-task VWM-B program training on reading-related EFs, reading skills, and reading comprehension. The VWM-B program was more effective than the VWM program in improving selective attention, cognitive inhibition, verbal WM capacity, information processing speed, naming ability, and lexical access, as evident by higher improvement in the scores obtained from BDS, TCT, PVFT, SCWI, SCWT, and all reading subtests of the NEMA (WRT, NWRT, CWT, and PDT). Additionally, the VWM-B program was found to be more efficient in enhancing measured reading skills, and the improved EFs were associated with improved reading skills. The study authors provided a comprehensive discussion of these findings.

The results revealed that the VWM-B is more effective than the VWM in boosting the scores on BDS, a valid scale for estimating the changes in the verbal WM capacity [92]. According to Baddeley’s WM model, the storage demands for a complex memory task like BDS are based on the efficient processing of phonological information, which is primarily facilitated by the central executive [31]. Therefore, BDS involves both the central executive and the phonological loop, thereby allowing for the interpretation of BDS results as a reflection of the central executive function [31,72]. If there is progress in BDS scores, it can be defined as improved cognitive inhibition, as the central executive component may be involved in inhibitory function [93]. Moreover, an increased WM capacity has been suggested to reflect improvements in processing speed and efficiency, which releases more resources to support storage [94]. Therefore, increased BDS scores may imply changes in cognitive inhibition, processing speed, and verbal WM capacity, which we will discuss further.

The link between reading disability and verbal WM dysfunction has been established [95]. An improved verbal WM capacity can lead to better reading skills [2,60,95]. Developing phonological awareness and reading ability relies on verbal WM capacity, and both verbal WM and phonological awareness are crucial for early literacy acquisition [95]. It has been suggested that both verbal WM and phonological awareness reflect a common phonological processing substrate [96]. In line with this, a positive correlation was discovered between increased verbal WM and enhanced phonemic awareness, tested by the phoneme deletion subtest of the NEMA.

As mentioned, the increased BDS scores may be interpreted as an improvement in cognitive inhibition and processing speed. Recent research has shown the correlation of WM with Stroop subtests (SCWT and SCWI) [97]. In many studies in different populations, the Stroop test was used to assess processing speed, cognitive inhibition, selective attention, and naming speed [2,60,98,99,100]. In the present study, improvement in these EFs was observed through changes in the SCWT and SCWI scores. Although other tests used in the present study (fluency test and TMT-A) have proven to be useful in evaluating processing speed, no significant change in TMT-A scores was observed after the intervention [98]. The results of Stroop tests support improvements in processing speed and inhibition, which have also been supported in previous studies [2,84,98,101]. Furthermore, the Stroop supports enhancements in selective attention and naming ability [2,60]. Overall, the significant change in Stroop variables in the present study could imply potential enhancements in processing speed, cognitive inhibition, selective attention, and naming ability. By supporting our findings, significant correlations were identified between Stroop SCWT and SCWI variables and the WCT subtest of the NEMA, indicating that these EFs were associated with an improvement in word recognition.

The study reveals a correlation between Stroop variables and TCT outcomes. The improvement in TCT scores suggests an improvement in reading comprehension. Processing speed and cognitive inhibition are critical for sufficient reading comprehension [102,103,104,105]. It is well established in the literature that text comprehension is linked to verbal WM function [106], and the current study found a correlation between TCT (text comprehension) and BDS (verbal WM). Previous research has demonstrated that dual-task cognitive–motor training can improve reading comprehension and reading skills [2,60], and our results align with that. The improved reading comprehension was strongly associated with word recognition (WCT) and was moderately associated with phonemic awareness (PDT) and decoding accuracy (NWRT).

According to the ANOVA results, the VWM-B led to a significant improvement in the PVFT scores. The improved phonemic verbal fluency showed a strong correlation with reading accuracy (WRT), word recognition (WCT), and phonemic awareness (PDT). Moreover, a moderate correlation was found between phonemic verbal fluency and verbal WM, which is supported by evidence [107]. Verbal fluency is a type of EF typically encompassing two categories: phonemic and semantic fluency [76,77]. Improved verbal fluency has been associated with an increased vocabulary size, faster lexical access, and cognitive inhibition [77]. Phonemic verbal fluency is particularly linked to the lexical access speed, which indicates an enhancement in processing speed [77]. Lexical access ability refers to the ability to retrieve the sound forms of words and grammatical representations from the mental lexicon [108].

Although it has been observed in several studies that dual-task trainings have benefits in various populations [109,110,111], the VWM-B program is the only dual-task program evaluated for individuals with DD [2,60]. This program is unique because it combines the maintenance sub-process of WM with balance-related complex movements [2]. The maintenance sub-process of the WM in the VWM-B program is performed within two states of balance—active and passive [2]. In the passive state, the cognitive task is prioritized, while in the active state, balance takes priority [2]. During dual tasks, the nervous system allocates more cognitive and attentional resources to the prioritized task, and the non-priority task receives less cognitive and attentional resources, leading to a decrease in performance [112]. The VWM-B program addresses both cognitive and balance disabilities of individuals with DD. Therefore, it provides sufficient opportunities for both cognitive and motor functions to be considered as a prioritized task and allows for further cognitive and attentional resource allocation in the nervous system [2]. Based on the automatization theory in DD, after several training sessions with the VWM-B program, balance-related movements would become automatized, leading to further resource allocation to the cognitive task [2,113]. Each prioritized cognitive and motor task in the VWM-B can lead to increased activation of specific cerebral or cerebellar regions. According to recent studies, the VWM-B program activates critical cerebral (the left fusiform gyrus) and cerebellar (Crus II) regions [2,60] which are essential for various reading skills [60,114,115,116,117,118,119,120,121]. Future studies can explore the possible association between the activation of cerebral and cerebellar regions induced by the VWM-B program and reading-related EFs.

In conclusion, the results support the hypothesis that the dual-task VWM-B program, compared to the single-task VWM program, can better enhance measured EFs related to reading, including selective attention, cognitive inhibition, verbal WM, processing speed, naming ability, and ability to lexical access in children with DD. These measured EFs are associated with improved reading skills and comprehension. It is important to note that phonemic awareness and decoding skills play a crucial role in word recognition [122,123,124]. It appears that VWM-B training boosts word recognition by promoting phonemic awareness and decoding skills. Consequently, improved word recognition leads to better reading ability. Previous reports have been supported by the current study’s findings, which show positive effects of the VWM-B on reading ability.

## 5. Limitations

In metropolitan regions like Tehran, the quality of educational services can vary between districts, even among different communities and countries [2]. It is important to evaluate and compare the effectiveness of the VWM-B on different functions in various communities and countries while considering the possible confounding effects of participants’ socioeconomic status [2]. On the other hand, using a large sample size would improve the statistical power of a study and can help in generalizing the results [60,86]. Although studies have shown positive effects of the VWM-B on various functions, these effects have only been reported in the short term [2,60]. Therefore, it is crucial to follow up on the long-term effectiveness of the VWM-B in the future. Thank you to an anonymous reviewer for suggesting the investigation of potential efficacy differences between the active and passive states of balance tasks in the VWM-B program, which could be considered for future research.

## Figures and Tables

**Figure 1 brainsci-14-00127-f001:**
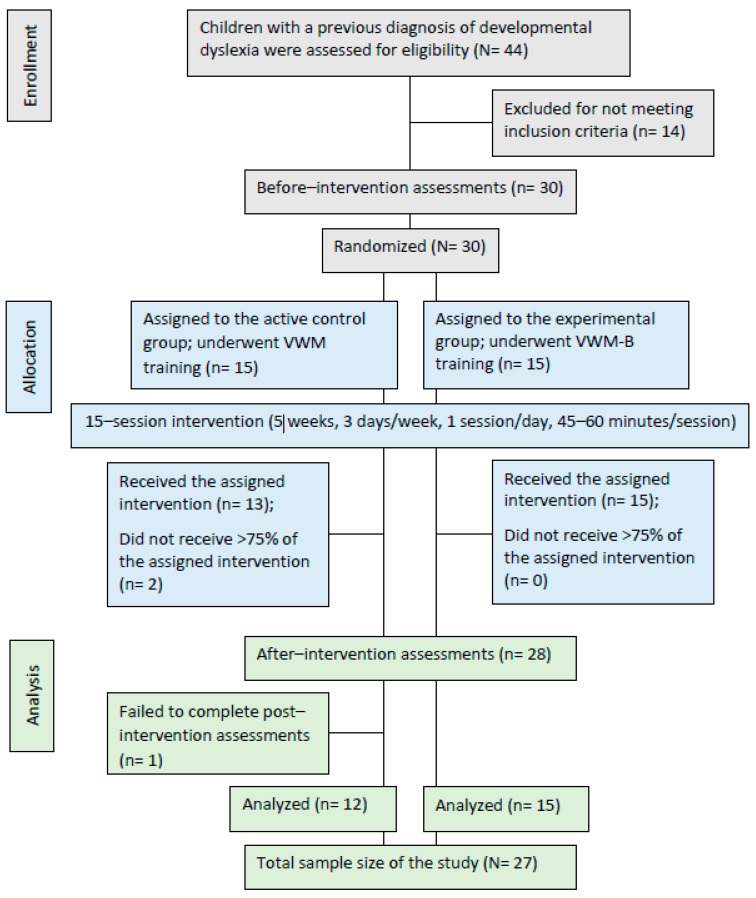
Illustration of the participants’ recruitment procedure.

**Figure 2 brainsci-14-00127-f002:**
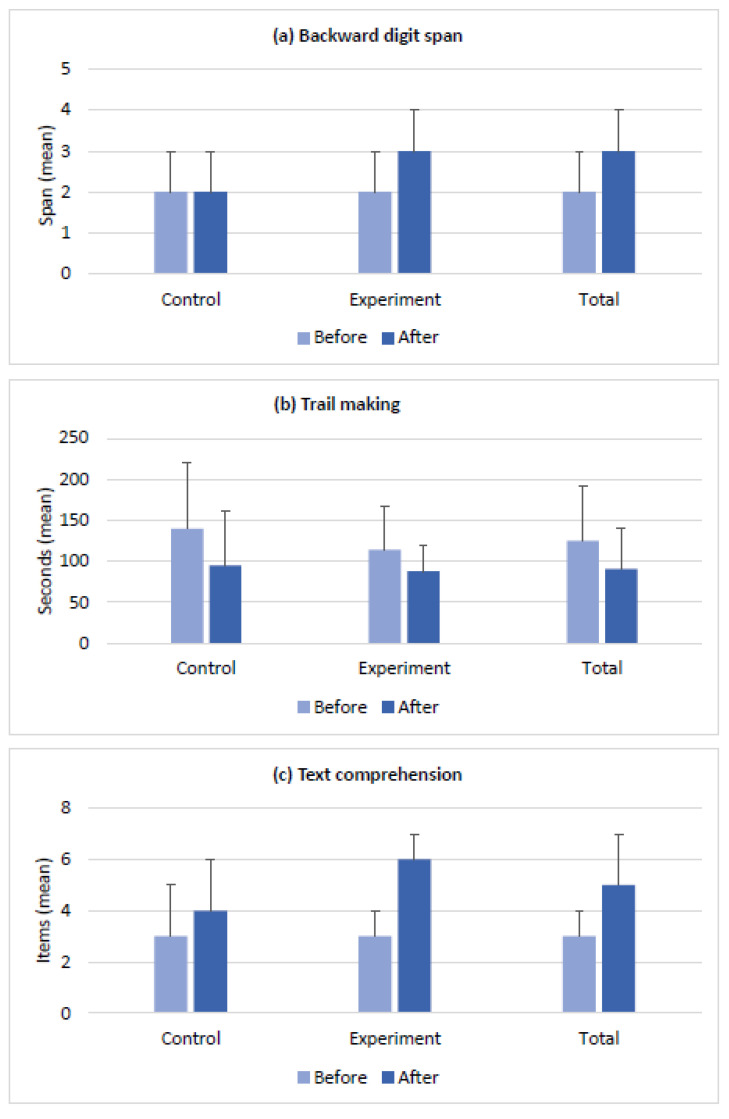
Clinical outcome measures (mean ± SD) for the control and experimental groups before and after the intervention. (**a**) The backward digit span was used to estimate the changes in the verbal working memory capacity and central executive function; (**b**) the trail-making test part A was used to indicate the changes in cognitive and visuomotor processing speed; and (**c**) the text comprehension test was used to demonstrate the change in the reading comprehension. The control and experimental groups had no significant differences in their baseline scores.

**Figure 3 brainsci-14-00127-f003:**
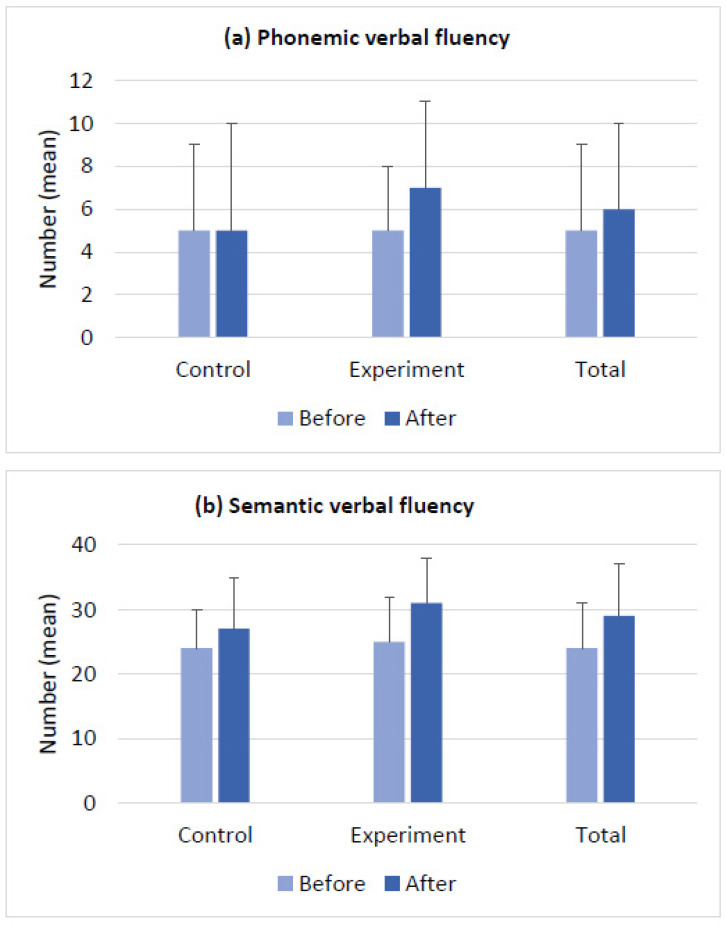
Verbal fluency outcomes (mean ± SD) for the control and experimental groups before and after the intervention. The verbal fluency test was used to estimate the changes in lexical access speed and working memory. (**a**) The phonemic verbal fluency test is designed to estimate the changes in lexical access speed; (**b**) the semantic verbal fluency test is designed to indicate the changes in the verbal working memory capacity. The control and experimental groups had no significant differences in their baseline scores.

**Figure 4 brainsci-14-00127-f004:**
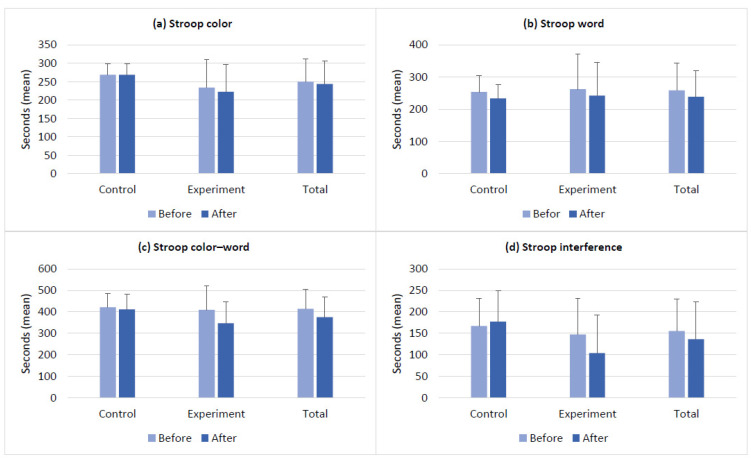
The Stroop test outcomes (mean ± SD) for the control and experimental groups before and after the intervention. (**a**) The Stroop color test was used to estimate the changes in color naming ability. (**b**) The Stroop word test was used to estimate the changes in word naming ability. (**c**) The Stroop color–word test, and (**d**) the Stroop interference was used to estimate the changes in processing speed, selective attention, inhibition, and naming ability. The control and experimental groups had no significant differences in their baseline scores.

**Figure 5 brainsci-14-00127-f005:**
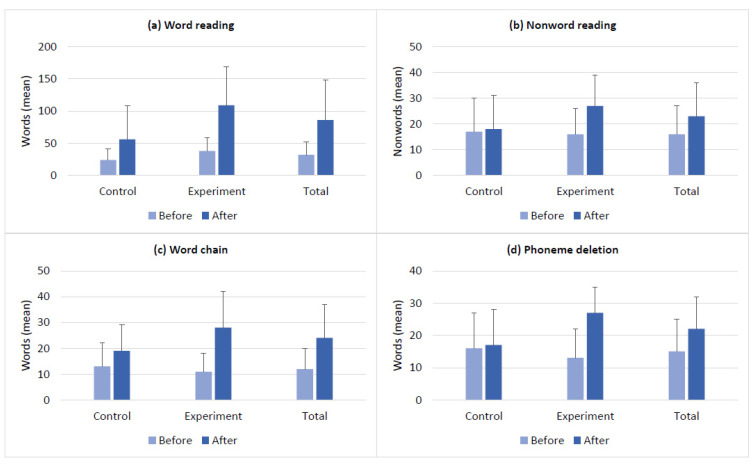
The NEMA reading subtests outcomes (mean ± SD) for the control and experimental groups before and after the intervention. (**a**) The word reading test was used to estimate the changes in word-reading accuracy and orthography. (**b**) The non-word reading test was used to estimate the changes in decoding accuracy. (**c**) The word chain test was used to estimate the changes in word recognition ability, and (**d**) the phoneme deletion test was used to estimate the changes in phonemic awareness. The control and experimental groups had no significant differences in their baseline scores.

**Table 1 brainsci-14-00127-t001:** Demographic characteristics.

Demography		Control (n = 12)	Experiment (n = 15)	Total (N = 27)	Group Differences (*p*-Value)
Mean (SD)					
Age (y)		9 (0.90)	8 (0.74)	9 (0.86)	u = 55.50 (0.065)
IQ (WISC-IV total score)		95 (6.89)	94 (7.99)	95 (1.42)	u = 82.00 (0.69)
Attention (CSI-4, total scores of 1 to 18 items)		5 (1.78)	3 (2.46)	4 (2.27)	u = 61.00 (0.146)
Frequency (%)					
Gender	Boy	4 (33.3)	3 (20.0)	7 (25.9)	χ^2^ = 0.617 (0.432)
	Girl	8 (66.7)	12 (80.0)	20 (74.1)	
Disability	Reading	3 (25.0)	2 (13.3)	5 (18.5)	χ^2^ = 4.78 (0.188)
	Reading and writing	4 (33.3)	10 (66.7)	14 (51.9)	
	Reading and math	0 (0)	1 (6.7)	1 (3.7)	
	Reading, writing, and math	5 (41.7)	2 (13.3)	7 (25.9)	
School grade	Second	4 (33.3)	11 (73.3)	15 (55.6)	χ^2^ = 4.47 (0.107)
	Third	3 (25.0)	2 (13.3)	5 (18.5)	
	Fourth	5 (41.7)	2 (13.3)	7 (25.9)	
Eyes condition	Normal	11 (91.7)	13 (86.7)	24 (88.9)	χ^2^ = 0.169 (0.681)
	Corrected	1 (8.3)	2 (13.3)	3 (11.1)	
Ears condition	Normal	12 (100)	14 (93.3)	26 (96.3)	χ^2^ = 0.831 (0.362)
	Corrected	0 (0)	1 (6.7)	1 (3.7)	

Note: No significant differences were found in terms of demographic data between children with dyslexia in the control and experiment groups. Abbreviations: IQ, intelligence quotient; WISC-IV, Wechsler intelligence scale for children—fourth edition; CSI-4, child symptoms inventory-4, total scores from items 1 to 18 of the parent checklist.

**Table 2 brainsci-14-00127-t002:** The mixed ANOVA analyses outcomes (N = 27).

Outcomes	Time Effects		Group Effects		Time × Group Interaction		
	*f* (1–25)	*p*-Value	*f* (1-25)	*p*-Value	*f* (1-25)	*p*-Value	Effect Size *η_p_^2^*
BDS	17.36	**<0.001**	1.90	0.180	9.25	**0.005**	0.27
TMT-A	21.43	**<0.001**	0.54	0.470	1.51	0.231	0.06
TCT	157.55	**<0.001**	1.81	0.191	14.40	**0.001**	0.37
PVFT	10.13	**0.004**	0.55	0.466	5.02	**0.034**	0.18
SVFT	18.07	**<0.001**	0.61	0.443	3.00	0.096	0.11
SCT	1.15	0.229	3.22	0.085	1.26	0.273	0.05
SWT	16.67	**<0.001**	0.08	0.785	0.00	0.991	0.00
SCWT	17.70	**<0.001**	1.20	0.283	9.15	0.006	0.27
SCWI	4.16	**0.052**	2.52	0.125	10.42	**0.003**	0.29
WRT	36.92	**<0.001**	6.04	**0.021**	5.11	**0.033**	0.17
NWRT	26.57	**<0.001**	1.28	0.268	8.76	**0.007**	0.26
CWT	52.85	**<0.001**	0.99	0.328	12.90	**0.002**	0.33
PDT	54.50	**<0.001**	3.25	0.083	12.82	**0.001**	0.34

Note: Bolded values indicate statistically significant *p*-values (*p* < 0.05). Abbreviations: BDS, backward digit span; TMT-A, trail-making test part A; TCT, text comprehension test; PVFT, phonemic verbal fluency test; SVFT, semantic verbal fluency test; SCT, Stroop color test; SWT, Stroop word test; SCWT, Stroop color–word test; SCWI, Stroop color–word interference; WRT, word reading test; NWRT, nonword reading test; CWT, chains word test; PDT, phoneme deletion test.

**Table 3 brainsci-14-00127-t003:** Pearson’s correlation between the reading skills and executive functions r (*p*-value).

Outcomes	WRT	NWRT	CWT	PDT
BDS	0.30 (0.129)	0.31 (0.115)	0.21 (0.304)	**0.56 ** (0.002)**
TMT-A	0.23 (0.242)	0.30 (0.130)	0.09 (0.650)	0.11 (0.575)
TCT	0.37 (0.057)	**0.42 * (0.029)**	**0.63 ** (<0.001)**	**0.47 * (0.015)**
PVFT	**0.55 ** (0.003)**	0.12 (0.552)	**0.65 ** (<0.001)**	**0.51 ** (0.007)**
SVFT	**0.48 * (0.011)**	0.17 (0.394)	**0.39 * (0.042)**	0.25 (0.208)
SCT	−0.03 (0.865)	0.11 (0.585)	−0.10 (0.639)	−0.01 (0.965)
SWT	0.25 (0.218)	0.11 (0.568)	0.21 (0.290)	−0.07 (0.721)
SCWT	−0.15 (0.467)	−0.07 (0.733)	**−0.40 * (0.037)**	−0.28 (0.159)
SCWI	−0.28 (0.164)	−0.13 (0.520)	**−0.53 ** (0.005)**	−0.26 (0.199)

Note: Bolded values indicate statistically significant *p*-values (*p* < 0.05). * Correlation is significant at the 0.05 level (2-tailed). ** Correlation is significant at the 0.01 level (2-tailed). Abbreviations: WRT, word reading test; NWRT, nonword reading test; CWT, chains word test; PDT, phoneme deletion test; BDS, backward digit span; TMT-A, trail-making test part A; TCT, text comprehension test; PVFT, phonemic verbal fluency test; SVFT, semantic verbal fluency test; SCT, Stroop color test; SWT, Stroop word test; SCWT, Stroop color–word test; SCWI, Stroop color–word interference.

## Data Availability

The current study data are available from the corresponding author on reasonable request.

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
