# Peer review of "Cognitive-Motor Training Improves Reading-Related Executive Functions: A Randomized Clinical Trial Study in Dyslexia"

_brainsci, 2024, doi:10.3390/brainsci14020127_

Round 1

Reviewer 1 Report

Comments and Suggestions for Authors

Authors in this study aimed to evaluate the short-term effects of the dual-task Verbal Working Memory-Balance (VWM-B) program training on reading-related executive functions. The study is a randomized clinical trial, in which participants (children 8-10yrs diagnosed with developmental dyslexia) were randomly divided in two groups (VWM/VWM-B) and tested before and after the program on a series of questionnaires and tasks. The VWM-B program was more effective than the control program in improving selective attention, cognitive inhibition, verbal WM capacity, information processing speed, naming ability and lexical access, in addition to be capable of enhancing reading skills and EFs associated with reading. In my opinion, the study manages to fulfil its aims, evaluating the effectiveness of the VWM-B program and highlighting important differences between the program addressing only Verbal Working Memory and the one also involving the Balance. I think the manuscript raises an important issue regarding the importance of considering also the body when addressing complex psychological deficits. Notwithstanding the overall good state of the manuscript, I would like to make some remarks/comments:

·       I think you should specify that the control group is an “active” control group (another similar intervention).

·       In your sample there is a relatively high dropout rate. Why? To assess the efficacy of a program the dropout rate is fairly important. I think you should detail more and, if the dropout rate is linked to the program, address ways to overcome this issue.

·       I would recommend addressing the very minor issue of “quasi-double blinded study” as soon you describe the study as a “double-blind”.

·       Is only one training trial enough to understand the task? Did you check that participants correctly performed the task? What happened to those that did not understand instructions or had trouble with balancing? Given your sample and the fact that this is probably the first time that participants are doing this kind of studies (and most certainly using a balancing platform) I think it is important to know if they are correctly performing the program.

·       The sentence: “The control and experiment groups had no significant difference in baseline scores” is important. I would put it also in the manuscript itself and not only in figures description. With regards to figures, I would also recommend an overall increase of the resolution and quality of figures and tables.

·       Did you analyse differences between active and passive state of the VWM-B? I think it would be quite important to see if these two “tasks” of the experimental group elicit different results or not.

·       In the introduction and/or discussion I would go more in detail about other possible explanations of dyslexic deficits (see for example Chiarenza, 1982; 1990; Chiarenza et al., 2014; Lachmann et al., 2005; Lachmann and van Leeuwen, 2007; Lachmann, 2018)

·       In the introduction and/or discussion it is important to give the larger context and mention other training involving the body/balance used with dyslexic participants (for a recent review about one of this training and the importance of an integrated perspective in dyslexia, see Pellegrino et al., 2023).

Reviewer 2 Report

Comments and Suggestions for Authors

In this manuscript, the authors investigate the short-term effects of cognitive-motor training (VWM-B) on reading-related executive functions (EFs) in children with developmental dyslexia (DD). The authors target various reading-related EFs and design a quasi-double-blinded randomized clinical trial to assess the impact of dual-task intervention compared with the normal single-task VWM program on reading skills. Participants are carefully selected and split into control and experimental groups to ensure homogeneity for potential confounding variables such as gender, IQ level, attention level, and socioeconomic status. Several measures are adopted to evaluate working memory (WM) capacity, selective attention, inhibition, information processing speed, and lexical access before and after the intervention.

Based on rigorous statistical tests, the dual-task VWM-B program significantly improves measured reading-related EFs, including selective attention, cognitive inhibition, verbal WM, processing speed, naming ability, and the ability to access lexical information in children with DD, compared with a single-task program. The measured EFs also exhibit a significant correlation with each other and with reading skills, indicating consistency in reading ability evaluation. This research is important as it provides clinical support for dual-task training as a more effective intervention for developmental dyslexia.

Comments:

1.       Abstract was missing

2.       In table 2 to table 5, since they are multiple comparisons, consider add corrections (such as Bonferroni correction) to the test results.

3.       Rephrase the statement in page 15 “By supporting our findings, significant correlations between Stroop SCWT and SCWI variables and WCT subtest of the NEMA, these EFs were associated with improvement of word recognition.” To “By supporting our findings, significant correlations were identified between Stroop SCWT and SCWI variables and the WCT subtest of the NEMA, indicating that these EFs were associated with an improvement in word recognition."

4.       I suggest that in the future, line numbers be added to the manuscript so the reviewer can easily refer to the context.

Comments on the Quality of English Language

See above

Author Response

Please see the attachment.  Editing the English undertaken by Fawcett.

Reviewer 3 Report

Comments and Suggestions for Authors

In this manuscript, the authors aimed to assess the effectiveness of dual-task VWM-B program training on reading-related EFs, reading skills, and reading comprehension. The authors found that its VWM-B program was more effective than the VWM program in improving selective attention, cognitive inhibition, verbal WM ability, information processing speed, naming ability, and lexical acquisition. In addition, the authors found that the VWM-B program was more effective in improving measured reading skills and that increases in EFs were associated with increases in reading skills. I still have some concerns about the current form of the manuscript. The authors need to address several aspects before this can be published as follows:

Main concerns:

1.    An abstract summarizing the article is missing from the manuscript.

2.    The methods section of the manuscript takes up a lot of space and needs to be streamlined.

3.    The results section of the manuscript focuses on describing variability and requires further analysis of the findings.

4.    The graphs obtained from the statistical analysis need to be labeled for significance.

Comments on the Quality of English Language

Moderate editing of English language required

Author Response

Please see the attachment.  English edited by Fawcett
